# Recent Advances in Photoacoustic Agents for Theranostic Applications

**DOI:** 10.3390/nano13040695

**Published:** 2023-02-10

**Authors:** Seongyi Han, Tsedendamba Ninjbadgar, Mijeong Kang, Chulhong Kim, Jeesu Kim

**Affiliations:** 1Departments of Cogno-Mechatronics Engineering and Optics & Mechatronics Engineering, Pusan National University, Busan 46241, Republic of Korea; 2Departments of Convergence IT Engineering, Mechanical Engineering, and Electrical Engineering, School of Interdisciplinary Bioscience and Bioengineering, Medical Device Innovation Center, Pohang University of Science and Technology (POSTECH), Pohang 37673, Republic of Korea

**Keywords:** photoacoustic imaging, theranostic agent, photothermal therapy, photodynamic therapy

## Abstract

Photoacoustic agents are widely used in various theranostic applications. By evaluating the biodistribution obtained from photoacoustic images, the effectiveness of theranostic agents in terms of their delivery efficiency and treatment responses can be analyzed. Through this study, we evaluate and summarize the recent advances in photoacoustic-guided phototherapy, particularly in photothermal and photodynamic therapy. This overview can guide the future directions for theranostic development. Because of the recent applications of photoacoustic imaging in clinical trials, theranostic agents with photoacoustic monitoring have the potential to be translated into the clinical world.

## 1. Introduction

Theragnosis techniques are widely studied as a patient management strategy that combines both biomedical imaging modalities and drug delivery systems [1]. In recent decades, nanoparticles with both diagnostic and therapeutic functions have been demonstrated using various biomedical imaging modalities [2,3,4]. Conventional imaging techniques, such as computed tomography (CT) [5], positron emission tomography (PET) [6], and magnetic resonance imaging (MRI) [7], are generally used to track nanoparticles. Although these imaging modalities have great advantages for the three-dimensional visualization of biological tissues, their high cost and large size may reduce their effectiveness as monitoring tools for theranostic nanoparticles. Particularly, the potential harm in using ionizing radiation or materials creates hurdles for clinical applications in humans.

Recently, optical imaging techniques have been used to monitor optically absorbing theranostic nanoparticles [8,9,10,11]. Compared with other imaging techniques, optical imaging is cost-efficient, easy to implement, capable of real-time imaging, and free from ionizing radiation. Moreover, optical imaging techniques can be used to monitor therapeutic agents for phototherapy, which typically includes photothermal therapy (PTT) [12,13,14] and photodynamic therapy (PDT) [15,16,17]. PTT is based on photothermal energy conversion, which is the energy transfer from absorbed light to heat. The generated heat increases the local temperature and destroys the target cells, typically tumor cells [18]. PDT uses photosensitizing agents to treat diseased cells. Cytotoxic reactive oxygen species (ROS) are generated by activating the photosensitizer using laser illumination [19]. Although optical imaging methods can track the treatment responses of phototherapy, their shallow imaging depth, owing to strong optical scattering in the biological tissues, significantly reduces their potential clinical translation.

Photoacoustic imaging (PAI) is a hybrid imaging technique that visualizes the optical absorption properties of biological tissues with ultrasound (US) resolution [20,21,22]. Compared to pure optical imaging techniques, PAI can significantly increase the imaging depth while maintaining a relatively good spatial resolution [23]. In particular, the resolution and imaging depth of PAI can be tuned between US and fluorescence imaging by adjusting the foci of the ultrasound and light sources (Figure 1) [24,25,26,27]. By inheriting the advantages of optical imaging techniques, PAI can also be used to monitor drug delivery, treatment responses, and agent assessment [28,29,30,31,32]. Moreover, using multispectral photoacoustic (PA) responses, the molecular functional information of biological tissues can be evaluated in vivo [33,34,35,36,37,38]. Therefore, in recent times, various theranostic agents have been developed and evaluated using the PAI technique [39,40,41,42,43,44].

In this paper, we review recent advances in PA agents for theranostic applications, especially in PTT and PDT. First, the principles and feasibility of PAI for monitoring theranostic agents were reviewed. The representative results of PA-guided phototherapy were then evaluated. This overview of PA agents provides a future direction for theranostic applications. By monitoring and evaluating the delivery efficiency and treatment responses, we can assess the potential feasibility of the agents to be translated into the clinical world.

## 2. Principles of Photoacoustic-Guided Phototherapy

PAI is based on the PA effect, in which light energy is converted into acoustic waves via thermoelastic expansion (Figure 2) [45]. When a short (typically a few nanoseconds) pulsed laser is irradiated, the light energy is absorbed by the target molecule according to its optical absorption characteristics. The electrons of the molecule are excited by the absorbed light energy and release the energy to return to their original ground state. One way to release energy is to use heat, which causes thermoelastic expansion. However, the expanded tissues contract rapidly after pulse duration. This rapid volume change generates vibrations that propagate in the form of acoustic waves, referred to as PA waves. The initial pressure of the PA waves can be described using the following equation:(1)$$P\propto \Gamma \left(T\right)\cdot \sigma \cdot \mu_{a}\cdot F$$

The initial pressure (P) of the generated PA waves is linearly proportional to four parameters: the Grüneisen parameter (Γ) that depends on the local temperature (T); the heat conversion efficiency (σ), which can be expressed as the residual of the quantum yield; the optical absorption coefficient ($\mu_{a}$), which varies with the wavelength of the excitation laser; and the optical fluence (F). Because PA signals are delivered in the form of US waves, conventional US transducers and image reconstruction systems can also be used for PA image generation. However, the information in the resulting PA images is the optical absorption characteristics of the biological tissue, which can be converted into molecular functional information, including hemoglobin oxygen saturation [46,47,48,49], hemodynamic responses [50,51,52], and agent uptake [53,54,55,56,57,58,59,60,61]. Therefore, various PA agents have been demonstrated to possess theranostic applications in vivo. Moreover, recent clinical trials of PAI systems exhibited the feasibility of the technique when translated into human studies [62,63,64,65,66,67,68,69,70,71], especially for cancer diagnosis [72,73,74,75,76].

PTT is a treatment method that causes cell necrosis by rapidly increasing the temperature of the target site. In PTT, near-infrared (NIR) light typically illuminates lesions after delivering light-absorbing agents that have a high photothermal conversion efficiency [77]. When heat is released, normal and diseased cells can be damaged. Therefore, PTT agents must be selectively delivered in lesions. To validate the targeting efficiency of these agents, noninvasive biomedical imaging techniques have been utilized. From this perspective, PAI is a good candidate for visualizing the biodistribution of agents because heat generation from light absorption generates PA waves.

PDT is another type of phototherapy that uses a photochemistry based approach. In PDT, light-activatable chemicals called photosensitizers are used to destroy diseased cells [78]. Photosensitizers are not cytotoxic until they are activated via illumination. After activation using a specific wavelength of light energy, the electrons in the photosensitizers are excited to a very unstable singlet state. Thereafter, excess energy is released in three ways: (1) radiative relaxation, which produces fluorescence; (2) nonradiative decay, which releases heat; and (3) internal conversion, which produces cytotoxic reactive oxygen species (ROS), such as singlet oxygen, superoxide radicals, hydroxyl radicals, and hydrogen peroxide [79]. The photosensitizer concentration at the target site is a key factor for improving the efficacy of PDT [80]. To evaluate the targeting efficiency, PAI has also been used to visualize the biodistribution of photosensitizers.

## 3. Photoacoustic Agents for Theragnosis

### 3.1. Photoacoustic Agents for Photothermal Therapy

Organic materials are widely used for contrast-enhanced PAI because of their good photostability and nontoxicity. Organic semiconductor polymers have been studied for PA-guided PTT by tuning their optical absorption characteristics. Chen et al. demonstrated NIR-absorbing semiconductor polymer nanoparticles for both contrast-enhanced PAI and PTT [81]. The developed polymers were synthesized with different molecular weights based on diketopyrrolopyrrole–dithiophenes (DPP-DTs), the sizes of which could be tuned by controlling the concentrations of the initial polymers (Figure 3a). Remarkably, the absorption peak of DDP-DT red-shifted with an increase in the particle size (Figure 3b). The results showed the potential of DDP-DTs for multispectral PAI with broadly tunable absorption peaks (630–811 nm) by adjusting the particle size. Moreover, DDP-DTs were PEGylated with carboxyl groups (PS-PEG-COOH) to function as PTT agents (Figure 3c). The encapsulated semiconductor polymer dots (DPP-DT Pdots) were delivered into H22 hepatocellular tumor-bearing mice, and PA images were obtained at an excitation wavelength of 808 nm (Figure 3d). The resulting images showed contrast enhancement (a 2.6-fold increase compared to the control group) in the tumor, verifying the accumulation of DPP-DT Pdots. The PTT efficiency was also evaluated by measuring the temperature increase under laser illumination at the tumor site (Figure 3e). The temperature was increased by 26 °C within 5 min in the DPP-DT-Pdot group, whereas insignificant temperature increases were observed in the control groups. Furthermore, when comparing the tumor volumes, the prognosis in the PTT-treated group with DPP-DT Pdots showed a significant therapeutic ability compared to the other groups (Figure 3f).

Phthalocyanine (Pc) is another type of organic material that has been widely used as a functional dye owing to its easily adjustable photochemical properties for strong absorption in the NIR region. While Pc dyes have mainly been studied as photosensitizers for PDT, Li et al. reported a Pc-based theranostic agent that showed a high PA signal and PTT ability [82]. They fabricated a supramolecular assembly using zinc Pc tetrasubstituted with 4-sulfonatophenoxy (PcS4) and 3-(N,N,N-trimethylammonium) phenoxy (PcN4) groups (Figure 4a). The developed agent exhibited strong PA signals at an excitation wavelength of ~700 nm (Figure 4b). To evaluate the in vivo contrast enhancement, PA images were obtained from 4T1 breast tumor xenografted mice after the intraperitoneal injection of the developed PcS4-PcN4 with a 200 μM concentration and 200 μL volume (Figure 4c). From the whole-body PA images, the signal enhancement at the tumor site was observed after the injection of the PcS4-PcN4 supramolecules. In contrast, no remarkable signal enhancement was observed in the mice injected with PcS4 or PcN4. The results showed that supramolecular interactions between PcS4 and PcN4 significantly enhanced the visibility of PA images. Moreover, the photothermal efficiency of PcS4-PcN4 was evaluated by monitoring the temperature under the 660 nm laser illumination (Figure 4d). Compared to the control group, the PcS4-PcN4-delivered site showed a temperature increase of approximately 25 °C. Accordingly, the relative tumor volume after treatment showed the treatment ability of PcS4-PcN4-assisted PTT, with a much slower size increase (Figure 4e).

Contrast agents that target specific tumors have also been studied. Fan et al. demonstrated PA-guided anticancer therapy using melanin nanoparticles (MNPs) coupled with poly-L-lysine (PLL) to target laryngeal squamous cell carcinoma (LSCC) [83]. They also added microRNAs (miRNAs), which can suppress tumor growth, to the MNP-functionalized PPL. The strategy in this study was as follows: (1) release miRNA to downregulate cancer-associated genes and (2) destroy tumor cells through PTT. To verify the treatment strategy, miRNA-bonded MNP-PLLs were delivered into Hep2 tumor xenografts via intratumoral injection. The PA images of the mice showed the accumulation of nanoparticles in the tumor region. Under 808 nm laser illuminations, the photothermal heat releases miRNAs by breaking molecular interactions between the nanocarriers and miRNAs. The infrared thermal images showed a temperature increase of 22.6 °C in the miRNA-bonded MNP-PLL-delivered mice, whereas a 3.0 °C increase was observed in the phosphate buffer saline (PBS)-delivered mice. Additionally, the released miRNAs suppressed tumor growth. The results showed a significant antitumor effect by combining PTT and gene therapy.

Metallic nanoparticles have received considerable attention because of their relative ease of tuning optical responses. Recently, Wang et al. introduced gold-based nanocomposites, named AuNSPHs, that exhibited high photothermal conversion efficiency [84]. AuNSPHs were based on gold nanostars (AuNSs) coated with polyaniline (PANI) using 1-dodecylmercaptan (DDT) as a linker (Figure 5a). To add targeting ability to tumors, hyaluronic acid coating was applied through electrostatic interactions with poly(diallyldimethylammonium chloride) (PDDA). The resulting AuNSPHs showed up to 78.6% photothermal conversion efficiency and strong PA signal generation at 850 nm (Figure 5b). For in vivo evaluation, PA images of 4T1 tumor-bearing mice were analyzed after the intravenous injection of AuNSPHs (Figure 5c). After 8 h of injection, the PA signals in the tumor region increased 2.3 times, showing an accumulation in AuNSPHs. Under 808 nm laser illumination, the temperature increased by 18.9 °C in the AuNSPH-delivered site, whereas only 3.7 °C was increased in the control group (Figure 5d). The relative tumor volume after PTT also showed the treatment efficacy of AuNSPHs (Figure 5e). Based on their outstanding photothermal conversion efficiency (78.6%), AuNSPHs can be potentially used as theranostic agents.

### 3.2. Photoacoustic Agents for Photodynamic Therapy

In recent decades, two-dimensional nanomaterials have been used to improve the efficiency of drug delivery systems. Lin et al. synthesized two-dimensional tellurium (Te) nanosheets that could produce ROS under light exposure, making them suitable for PA-guided PDT [85]. The Te nanosheets were initially synthesized using the sonication of raw Te powder and then functionalized using glutathione (GHS) conjugations, which made the resulting nanosheets (TeNS@GHS) stable and biocompatible. The ROS generation of TeNS@GHS was verified by measuring the electron spin resonance signals under 670 nm light illumination. To evaluate the feasibility of the PA-guided therapy, in vivo PA images were obtained after the intratumoral injection of TeNS@GHSs into HepG2 tumor-bearing mice. After injection, the accumulation of the TeNS@GHSs showed a ~21-fold increase in the PA signals at the tumor site. Additionally, the relative tumor volume was monitored after illumination (670 nm, 160 mW/cm^2^) for 10 min. The results showed significantly suppressed tumor growth in the treated group compared to that in the other groups.

Zhang et al. reported mesoporous platinum (mPt) nanoplatforms that efficiently catalyzed the conversion of hydrogen peroxide to oxygen and generated ROS under laser illumination [86]. They connected the polyethylene glycol-modified photosensitizer chlorin e6 (PEG-Ce6) onto mPt nanoparticles to construct PDT nanoplatforms named Pt@PEG-Ce6. The developed nanoplatforms possessed efficient PDT ability by releasing oxygen in a hypoxic tumor environment. The multispectral in vivo PA images of 4T1 xenografts in mice, acquired after the intravenous injection of Pt@PEG-Ce6s, verified the oxygen release from an increase in the oxygen saturation level. The monitored tumor volume after treatment evaluated the efficacy of PDT, showing suppressed tumor growth in the treated group compared to that in the control group.

Xavierselwan et al. also demonstrated theranostic perfluoropentane (PFP) nanodroplets that can carry three agents, i.e., benzoporphyrin derivative photosensitizer (BPD), indocyanine green (ICG), and oxygen (Figure 6a) [87]. Under laser illumination, ICG absorbs light energy and initiates the vaporization of PFP nanodroplets, which release oxygen. Therefore, the developed droplets can be used as light-triggering drug delivery platforms. To test the ability of PDT, the treatment laser beam (690 nm, 150 mW/cm^2^) was irradiated on FaDu xenografts in mice after the tail vein injection of the PFP droplets. Multispectral PAI verified the oxygen release at the tumor site by measuring a ~9.1-fold increase in the oxygen saturation level (Figure 6b). After PDT, the tumor growth was successfully suppressed by the nanodroplets (Figure 6c). Tumor prognosis showed that PFP containing BPD and ICG at a ratio of 1:1 efficiently performed PDT.

### 3.3. Photoacoustic Agents for Combined Photothermal and Photodynamic Therapy

Studies have attempted to perform both PTT and PDT using a single agent. Tang et al. developed an organic semiconducting nanodroplet for PAI-guided PTT and PDT (Figure 7a) [88]. The nanodroplet PS-PDI-PAnD was synthesized using encapsulating a ZnF_16_Pc photosensitizer with perfluorocarbons (PFCs) stabilized via light-absorbing perylene diimides (PDIs). Under laser illumination, PDIs effectively absorb energy and possess a high photothermal conversion efficiency. Moreover, the temperature increase activates the vaporization of PFC molecules, releasing the oxygen that can be consumed by the photosensitizer to produce cytotoxic singlet oxygen for PDT. Furthermore, vaporized PFCs generate bubbles that can be visualized using US imaging (USI). To validate the feasibility of image-guided therapy, these agents were intravenously injected into U87MG tumor-bearing mice. The in vivo tumor images showed contrast enhancement in both PA and US images owing to the accumulation of PS-PDI-PAnDs (Figure 7b). After the ten-minute irradiation laser (671 nm, 0.5 W/cm^2^), the relative tumor volume was significantly suppressed in the PS-PDI-PAnD-delivered group, whereas other groups failed to suppress the tumor growth (Figure 7c). Nanoparticles (PDI-PAnP), which were not encapsulated in the photosensitizers, also showed therapeutic effects owing to the photothermal conversion of the PDI shells. However, the absence of PDT resulted in a lower therapeutic effect compared to the PS-PDI-PAnD-delivered group.

Wang et al. developed spiky supraparticles (SPs) that showed good therapeutic effects with multimodal imaging capabilities [89]. They synthesized SPs via a simple seed-mediated growth method, using gold-coated iron oxide (Fe_3_O_4_@Au) as the seed (Figure 7d). Fe_3_O_4_@Au SP demonstrated great feasibility as a multimodal imaging agent for CT, MRI, and PAI (Figure 7e). Under the laser illumination (808 nm, 0.5 W/cm^2^), photothermal conversion was successfully monitored with a conversion efficiency of 31% and a temperature increase of 65.7 °C. Moreover, ROS generation was verified by measuring fluorescence signals, which is evidence of the generation of singlet oxygen, after laser illumination with the same wavelength (808 nm). The therapeutic ability of Fe_3_O_4_@Au SPs was evaluated in HeLa cancer-xenografted mice. After the intravenous injection of the agents, the treatment laser (808 nm, 0.5 W/cm^2^) was irradiated for 5 min. The resulting relative tumor volume showed that the Fe_3_O_4_@Au SPs succeeded in curing the tumor, whereas other controls failed to suppress tumor growth (Figure 7f).

## 4. Discussion

Recent studies on PAI-guided phototherapy are summarized in Table 1. Because both PAI and phototherapy can be derived from the optical absorption of nanomaterials, PAI can be used as a monitoring tool with minimal modification of the therapeutic agents. By monitoring a series of PA images, we can monitor the delivery of agents before treatment, which can assess their targeting ability for efficient treatment. In addition to the contrast agents discussed in this review, various NIR dyes can be utilized for synthesizing therapeutic agents to monitor the targeting efficiency. Particularly, fluorescent dyes [90,91,92] with an adequate quantum yield can expand their application area to PAI [23]. This overview of therapeutic agents can guide future directions for the development of image-guided therapy. Based on recent clinical trials of PAI, theranostic agents with PA monitoring would have great potential for translation into the clinical world.

To improve the treatment efficiency, the optimal synthesis of both PTT and PDT agents is required. In PTT, increasing the photothermal conversion efficiency is key for efficient treatment. There have been studies conducted to determine the optimal size of nanomaterials and apply various shapes of nanomaterials, including nanostars and spiky supraparticles. In PDT, overcoming the hypoxic conditions in the tumor region can improve the treatment efficiency because the generation of ROS typically consumes oxygen. Nanodroplets that can deliver oxygen in addition to photosensitizers are a good option to overcome the hypoxic condition of the tumor site. Under laser irradiation, oxygen is released, and ROS generation is activated. Further, computational studies for evaluating state dynamics can be applied to design efficient therapeutic agents [93].

Recently, agents that absorb lasers in the second near-infrared (NIR-II) region (1000–1700 nm) have been widely studied for PA contrast. In this region, the photon scattering in biological tissues is significantly less than that in the first near-infrared (NIR-I) region (650–1000 nm), which is mainly presented in this paper. Although PDT may be limited due to the lower efficiency of energy transfer to oxygen in NIR-II compared to the visible and NIR-I ranges, the less scattering of light is highly beneficial in biomedical studies as it enhances the penetration depth [94], allowing higher laser energy delivery to the skin [95] and stronger PA signal generation. Therefore, therapeutic agents that absorb in the NIR-II region can be used for drug delivery to deep tissues [96,97,98].

Advances in PAI systems may also be beneficial for biomedical applications of therapeutic agents. To visualize the biodistribution of agents, whole-body PAI systems equipped with multiple or tunable laser wavelengths have been widely developed [22]. Recently, three-dimensional PA computed tomography (PACT) systems have shown great potential for monitoring the biodistribution of small animals in vivo [26]. In addition, advanced deep-learning algorithms have been reported to achieve better image quality [99,100,101,102,103,104]. The continuous improvement of the PAI system can potentially translate the PAI-guided phototherapy technique to clinical applications.

In addition to the monitoring of the delivery, techniques to control the release of drugs have also been studied [105,106,107]. The key to a controlled release is the triggering mechanism caused by external stimuli or environmental change. The representative external stimuli related to PAI are light [108], temperature [109], and US waves [110]. Monitoring those drugs using PAI can improve the efficiency of the release by determining the optimal time for triggering. In recent years, the controlled release of hydrogen (H_2_) gases has been studied [111,112,113], showing synergetic treatment with phototherapies. The difference in the biological environments such as pH level [114], enzymatic reactions [115], and redox reactions [116] in the diseased region also can be used to trigger drug release. When the agents meet the environmental change, the shape of the agents is transformed to release the drugs, causing shifted optical absorption characteristics. In this perspective, PAI can noninvasively verify the drug release in the target tissues in vivo [117,118,119].

## Figures and Tables

**Figure 1 nanomaterials-13-00695-f001:**
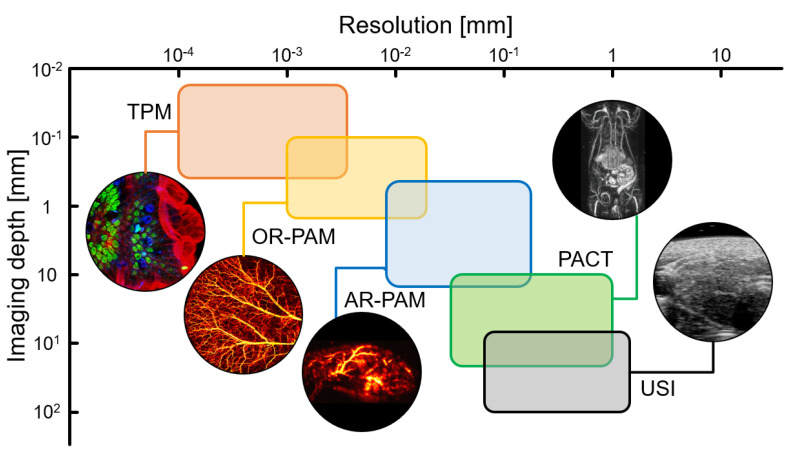
Resolution and imaging depth of the imaging modalities. TPM, two−photon fluorescence microscopy; OR−PAM, optical−resolution photoacoustic microscopy; AR−PAM, acoustic-resolution photoacoustic microscopy; PACT, photoacoustic computed tomography; USI, ultrasound imaging; The images are reproduced with permission from Refs. [24,25,26,27].

**Figure 2 nanomaterials-13-00695-f002:**
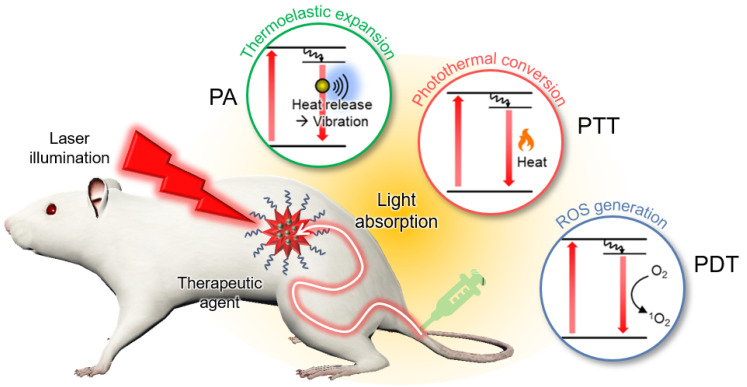
Schematic illustration of PA-guided phototherapy. PA, photoacoustic; PTT, photothermal therapy; PDT, photodynamic therapy; ROS, reactive oxygen species.

**Figure 3 nanomaterials-13-00695-f003:**
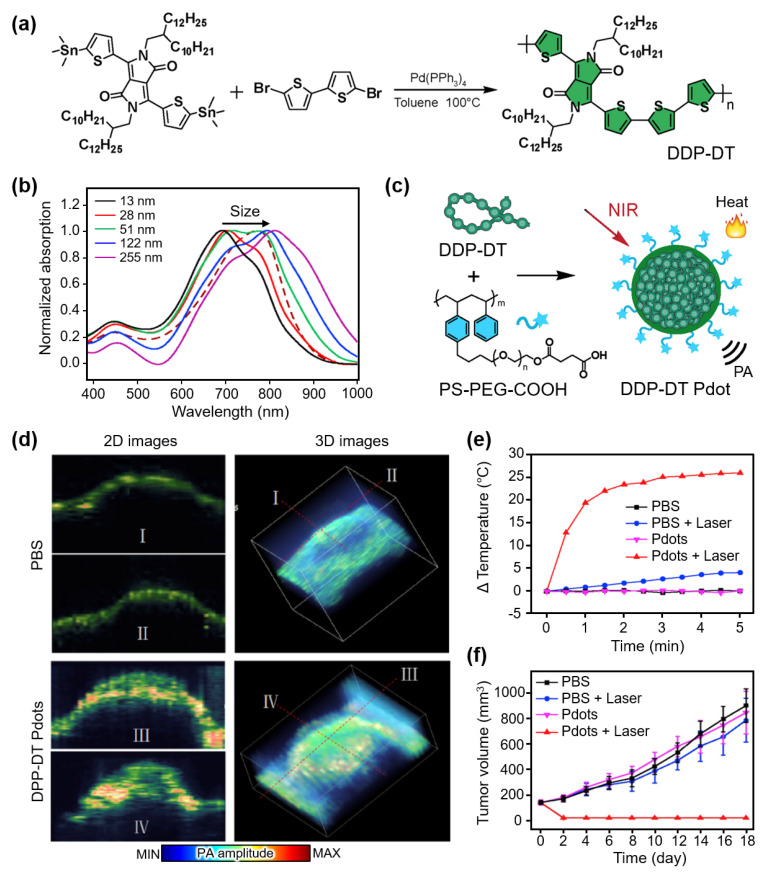
(**a**) Synthesis of DPP-DT. (**b**) Absorption spectra of DPP-DTs with varying particle sizes. (**c**) Schematic illustration for photothermal and PA wave generation of DPP-DT Pdots. (**d**) In vivo PA images of tumor-bearing mice after injection of saline and DPP-DT Pdots. (**e**) Temperature increase at the tumor site under laser illumination. (**f**) Averaged tumor volume after treatment. DPP-DT, diketopyrrolopyrrole-dithiophenes; PA, photoacoustic; Pdot, semiconductor polymer dot; PBS, phosphate buffered saline. The images are reproduced with permission from Ref. [81].

**Figure 4 nanomaterials-13-00695-f004:**
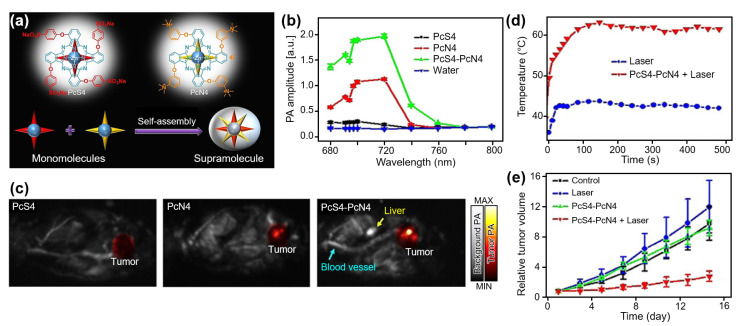
(**a**) Schematic illustration of supramolecular assembly. (**b**) PA amplitudes of 25 uM PcS4, PcN4, PcS4-PcN4 supramolecule, and water. (**c**) In vivo PA images of tumor-bearing mice at 72 h after injection. (**d**) Temperature increase at the tumor site under laser irradiation. (**e**) Relative tumor volume after treatment. PcS4, zinc Pc tetrasubstituted with 4-sulfonatophenoxy groups; PcN4, zinc Pc tetrasubstituted with 3-(N,N,N-trimethylammonium) phenoxy groups; PA, photoacoustic. The images are reproduced with permission from Ref. [82].

**Figure 5 nanomaterials-13-00695-f005:**
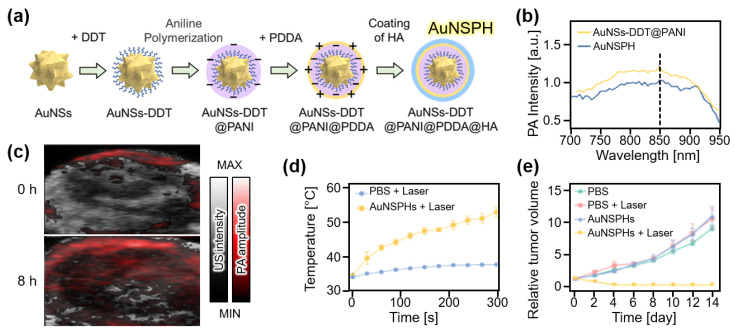
(**a**) Schematic illustration of the synthesis of AuNSPHs. (**b**) PA amplitude of AuNSPHs. (**c**) In vivo overlaid PA and US images of tumor region in mice at 0 and 8 h after injection of AuNSPHs. (**d**) Temperature increase at the tumor region under the illumination of the laser. (**e**) Relative tumor volume after treatment. AuNS, gold nanostar; DDT, 1-dodecylmercaptan; PANI, polyaniline; PDDA, poly(diallyldimethylammonium chloride); HA, hyaluronic acid; AuNSPH, AuNSs-DDT@PANI@PDDA@HA; PA, photoacoustic; US, ultrasound; PBS, phosphate buffered saline. The images are reproduced with permission from Ref. [84].

**Figure 6 nanomaterials-13-00695-f006:**
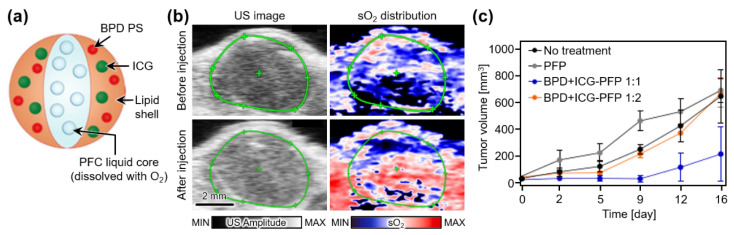
(**a**) Schematic illustration of theranostic PFP nanodroplets. (**b**) In vivo US images and corresponding sO_2_ distribution map at tumor region in mice before and after injection of PFP nanodroplets. (**c**) Measured tumor volume after treatment. PFP, perfluoropentane; BPD, benzoporphyrin derivative photosensitizer; ICG, indocyanine green; PFC, perfluorocarbon; US, ultrasound; sO_2_, oxygen saturation level. The images are reproduced with permission from Ref. [87].

**Figure 7 nanomaterials-13-00695-f007:**
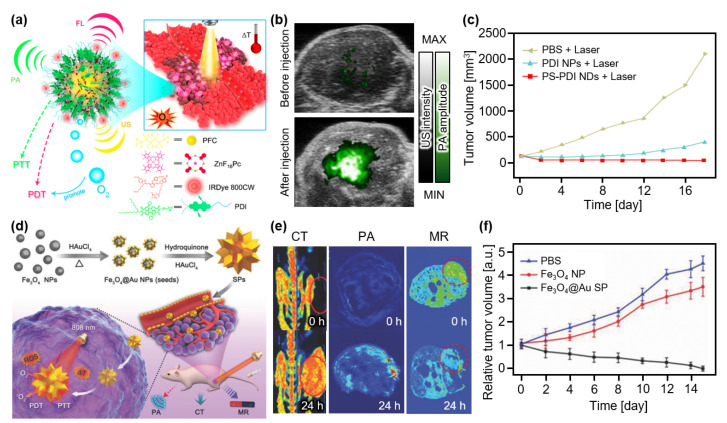
(**a**) Schematic illustration of phototherapy with the PS-PDI NDs. (**b**) Overlaid PA and US images of tumor region in mice before and after injection of the PS-PDI NDs. (**c**) Measured tumor volume after treatment. (**a**–**d**) Schematic illustration of the Fe_3_O_4_@Au SPs. (**e**) In vivo CT, PA, and MR images of tumor-bearing mice at 0 and 24 h after injection of the Fe_3_O_4_@Au SPs. (**f**) Relative tumor volume after treatment. Fe_3_O_4_@Au, gold-coated iron oxide; NP, nanoparticle; SP, supraparticle; PA, photoacoustic; CT, X-ray computed tomography; MR, magnetic resonance; US, ultrasound; PTT, photothermal therapy; PDT, photodynamic therapy; PBS, phosphate buffered saline; PS, photosensitizer; PDI, perylene diimides; ND, nanodroplet. The images are reproduced with permission from Refs. [88,89].

**Table 1 nanomaterials-13-00695-t001:** Summary of PAI-guided phototherapy. $\lambda_{PA}$, wavelength for PAI; $\lambda_{T}$, wavelength for treatment; PTT, photothermal therapy; PDT, photodynamic therapy; SPN, semiconductor polymer nanoparticle; NP, nanoparticle; DDP-DT, diketopyrrolopyrrole-dithiophene; Pc, phthalocyanine; SP, supramolecule; Au, gold; PLL, poly-L-lysine; MNP, melanin nanoparticle; Te, tellurium; mPt, mesoporous platinum; PFP, perfluoropentane; Fe_3_O_4_, iron oxide; PAI, photoacoustic imaging; USI, ultrasound imaging; FLI, fluorescence imaging; MRI, magnetic resonance imaging; CT, X-ray computed tomography.

Treatment Methods	Type of Materials	Characteristics	Application (Cell Line)	Imaging Methods	Laser			Ref.
					$\lambda_{PA}$	$\lambda_{T}$	Power [W/cm^2^]	
PTT	DPP-DT based SPN	Organic materials with good photostability and nontoxicity	H22	PAI	700	808	0.5	[81]
	Zinc Pc tetrasubstituted SP	Organic materials with strong absorption in the NIR region	4T1	PAI	700	660	0.6	[82]
	PLL coupled MNP	Organic NPs targeted to specific tumor	Hep2	PAI	680	808	1.5	[83]
	Au nanocomposite	Metallic NPs with efficient photothermal conversion	4T1	PAI, USI	850	808	1	[84]
PDT	Two-dimensional Te nanosheet	Te nanostructure for image-guided therapy	Hep2	PAI	-	670	0.16	[85]
	mPt nanoplatform	Metallic oxygen-releasing agents for efficient therapy	4T1	PAI, USI, CT	532	660	1	[86]
	PFP nanodroplet	Vaporizable nanodroplets release oxygen for efficient therapy	FaDu	PAI, USI	800	690	0.04	[87]
PDT/PTT	Semiconducting nanodroplet	Organic agent for multimodal image-guided therapy	U87MG	PAI, USI, FLI	808	671	0.5	[88]
	Au-coated Fe_3_O_4_ SP	Metallic agent for multimodal image-guided therapy	HeLa	PAI, MRI, CT	808	808	0.5	[89]

## Data Availability

Not applicable.

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
