# Peer review of "Recent Advances in Photoacoustic Agents for Theranostic Applications"

_nanomaterials, 2023, doi:10.3390/nano13040695_

Round 1
Reviewer 1 Report
I carefully read the manuscript review article by Seongyi Han et. al. entitled “Recent Advances in Photoacoustic Agents for Theragnostic Applications”.
In my opinion, the work is done at a good scientific level, written in an accessible language and easy enough for the reader to understand.
However, there are a number of shortcomings in the work that must be corrected before publication in the journal Nanomaterials.
Major remarks:
1) First of all, the term "Theragnostic" is confusing. In my opinion, it is better to write "Theranostic" - this term is 15 times more often used in this subject.
2) In the Introduction chapter (for example, line 50), the authors did not pay enough attention to the advantages of the PAI method over fluorescent or optical imaging. I would like to see figures comparing the depth of sounding at a particular wavelength, spatial resolution, etc., in order to understand where it is more profitable to use PAI.
3) The Discussion chapter discusses the advantages of the NIR-II range, but does not say about the limitations of this range for PDT. The efficiency of energy transfer to oxygen in this range is much lower than in the visible and NIR-I ranges, since the energy of the 3O2-1O2 transition is approximately 1260 nm.
4) Line 32: "their high cost and large size may degrade the efficiency of theragnostic nanoparticles". It is not clear how the large size of devices can affect the effectiveness of nanoparticles for theranostics?
Minor remarks:
1) The terms in vivo, in vitro, et. al. and the like should be in italics.
2) Line 252: "Tumor prognosis showed that PFP 251 containing BPD and ICG at a ratio of 1:2 efficiently performed PDT." Judging by Figure 5c, particles with a ratio of 1:1 had the highest efficiency.
I believe that with the correction of these shortcomings, the article can be published in the journal Nanomaterials.
Author Response
Thank you for your comments. Please find our responses in the attached file.

Reviewer 2 Report
Comments:
Photoacoustic technology is an emerging biomedical imaging modality that images optical absorbers in tissue through an acoustic detector. Photoacoustic imaging allows for high resolution and deep penetration. In recent years, tremendous developments have been made in both instrumentation and imaging agents. By assessing the biodistribution of photoacoustic images, the effectiveness of a drug in terms of delivery efficiency and therapeutic response can be analyzed.
In this work, the authors evaluate and summarize the latest advances in photoacoustic-guided phototherapy, especially in PDT/PTT. A direction for the development of photoacoustic-guided precision phototherapy is provided. Overall, the whole process is organized, but not very comprehensive. I recommend accepting the manuscript after revisions.
1. Authors should check the use of uniform fonts, such as "in vivo".
2. There are redundant punctuation marks in the labeling of Figure 3 (166-167).
3. The resolution of Figure 1 should be improved.
4. The material characteristics and categories of photoacoustic agents with visualization function should be summarized more abundantly in recent years, and the current summary of various materials is relatively scattered and not comprehensive enough. It is hoped that certain design ideas could be given to help researchers to design and develop novel synergistic therapeutic materials.
5. More theoretical computational studies such as excited state dynamics relevant to detection and treatment could be found in discussions such as Accounts of Chemical Research 2012, 45, 404.
6. Does the author have some unique insights into the controlled release scheme of single-platform novel materials in collaboration with PA/PDT/PTT therapy?
7. Near-infrared (NIR) imaging provides an innovative strategy for non-radioactive detection with high spatio-temporal resolution. In addition to cyanine dyes, other classes of NIR dyes are available for imaging. If the authors are interested in researching NIR imaging materials, the authors could refer to as discussed in Sensors and Actuators: B. Chemical 2023, 380, 133320; Analytica Chimica Acta 2023, 1241, 340778; SmartMat 2021, 2, 554.
8. In addition to the research work on the ability to monitor targeted drug delivery using photoacoustic agents, there is also research work on the use of inorganic materials for controllable release of gases such as H2 for the treatment of cancer cells that could be referenced.
Author Response

(The authors gave the same response as above.)

Round 2
Reviewer 1 Report
All comments and suggestions were taken into account and corrected
Reviewer 2 Report
The authors have made more careful revisions and I think the manuscript can be accepted.